# Fully Automated Photoplethysmography-Based Wearable Atrial Fibrillation Screening in a Hospital Setting

**DOI:** 10.3390/diagnostics15101233

**Published:** 2025-05-14

**Authors:** Khaled Abdelhamid, Pamela Reissenberger, Diana Piper, Nicole Koenig, Bianca Hoelz, Julia Schlaepfer, Simone Gysler, Helena McCullough, Sebastian Ramin-Wright, Anna-Lena Gabathuler, Jahnvi Khandpur, Milene Meier, Jens Eckstein

**Affiliations:** 1Department of Internal Medicine, University Hospital Basel, 4031 Basel, Switzerlandjens.eckstein@usb.ch (J.E.); 2Preventicus GmbH, 07743 Jena, Germany; 3Innovation Management, Department of D&ICT, University Hospital Basel, 4031 Basel, Switzerland

**Keywords:** wearable, atrial fibrillation, photoplethysmography, screening

## Abstract

**Background/Objectives**: Atrial fibrillation (AF) remains a major risk factor for stroke. It is often asymptomatic and paroxysmal, making it difficult to detect with conventional electrocardiography (ECG). While photoplethysmography (PPG)-based devices like smartwatches have demonstrated efficacy in detecting AF, they are rarely integrated into hospital infrastructure. The study aimed to establish a seamless system for real-time AF screening in hospitalized high-risk patients using a wrist-worn PPG device integrated into a hospital’s data infrastructure. **Methods**: In this investigator-initiated prospective clinical trial conducted at the University Hospital Basel, patients with a CHA_2_DS_2_-VASc score ≥ 2 and no history of AF received a wristband equipped with a PPG sensor for continuous monitoring during their hospital stay. The PPG data were automatically transmitted, analyzed, stored, and visualized. Upon detection of an absolute arrhythmia (AA) in the PPG signal, a Holter ECG was administered. **Results**: The analysis encompassed 346 patients (mean age 72 ± 10 years, 175 females (50.6%), mean CHA_2_DS_2_-VASc score 3.5 ± 1.3)). The mean monitoring duration was 4.3 ± 4.4 days. AA in the PPG signal was detected in twelve patients (3.5%, CI: 1.5–5.4%), with most cases identified within 24 h (*p* = 0.004). There was a 1.3 times higher AA burden during the nighttime compared to daytime (*p* = 0.03). Compliance was high (304/346, 87.9%). No instances of AF were confirmed in the nine patients undergoing Holter ECG. **Conclusions**: This study successfully pioneered an automated infrastructure for AF screening in hospitalized patients through the use of wrist-worn PPG devices. This implementation allowed for real-time data visualization and intervention in the form of a Holter ECG. The high compliance and early AA detection achieved in this study underscore the potential and relevance of this novel infrastructure in clinical practice.

## 1. Introduction

Atrial fibrillation (AF) is the most common cardiac arrhythmia and its prevalence is expected to rise in the coming years [1]. It has a significant economic impact due to its association with an increased all-cause mortality and morbidity, particularly a five-fold higher risk of stroke [2,3,4]. Early diagnosis of AF is crucial for preventing strokes and other thromboembolic events [5]. AF can be asymptomatic in up to one third of patients and often remains undetected until the first thromboembolic event occurs [6,7]. Furthermore, even when intermittent screening with conventional ECG is performed, episodes can be missed due to the paroxysmal nature of AF. The significance of early detection of subclinical atrial fibrillation is underscored by recent findings indicating that anticoagulant therapy can effectively reduce the risk of stroke or systemic embolism in this population [8]. This highlights the critical need for accurate and continuous monitoring systems to identify subclinical AF and facilitate timely therapeutic interventions.

Continuous ECG monitoring with implantable loop recorders and extended-duration Holter ECGs is increasingly used to screen for AF. While multiple studies have showed that these devices are highly effective at detecting AF in selected patients [9,10,11], their high cost and invasive nature limit their widespread use among the broader population.

Recent advances in wearable technology have enabled the development of non-invasive devices for AF screening. These include photoplethysmography (PPG)-based devices such as smartwatches, smartphone-based technology, and single-lead ECG technologies like smart chest patches. PPG-based devices enable indirect monitoring of the heart rate and have been demonstrated to accurately detect AF [12,13,14,15,16]. Large-scale studies including the Apple Heart Study and the Fitbit Heart Study have shown the feasibility of PPG-based AF detection [16,17]. Smartphone-based PPG recording using the built-in camera may be easily applicable because it does not require additional measurement devices, but the screening is not continuous. In contrast, wrist-worn wearables with integrated PPG sensors are increasingly available and can be worn for extended periods, resulting in near-continuous AF screening. Accordingly, the current guidelines of the European Society of Cardiology (ESC) have integrated the use of PPG sensors in the management and diagnosis of AF [6].

A recent study compared the diagnostic accuracy of a wrist-worn PPG device and single-lead ECG with the gold-standard Holter ECG, showing excellent sensitivity and specificity in both modalities, highlighting the clinical potential of such technologies [18]. A meta-analysis looking at 15 studies comparing ECG smart chest patches and PPG smartwatches found that both modalities had an excellent diagnostic performance in AF detection with pooled sensitivities of 96.1% and 97.4% and specificities of 97.5% and 96.6%, respectively [19]. The accurate detection of AF from PPG signals requires sophisticated algorithms to rightfully differentiate true arrhythmias from noise and other artefacts arising from motion or physiological variations [20,21]. Several algorithms have been developed and validated for AF detection using PPG. Table 1 provides a comparison of key characteristics and performance metrics of commonly used algorithms.

Various PPG-based algorithms demonstrate high sensitivity, specificity and positive predictive value. It is important to note that direct comparisons between PPG-based algorithms can be influenced by the study population and the type of PPG device investigated. The Preventicus Heartbeats Algorithm, used in this study, has been shown to have a sensitivity of 93.7% and a specificity of 98.2% [13], making it an ideal choice due to its excellent overall diagnostic performance across all key parameters.

Despite the progress made in this area, the integration of PPG-based wearables into hospital infrastructure remains rare, even though hospitalized patients represent a high-risk group for AF. Most studies to date have focused on an ambulatory population. This study focuses on the development and deployment of an automated system for continuous AF screening in a hospital setting. The goal was to design an IT infrastructure that enables the fully automated acquisition, processing, and real-time visualization of PPG data from wrist-worn wearable devices, allowing for their seamless integration into clinical workflows. By leveraging automated data analysis and visualization, this system aims to facilitate real-time AF screening for hospitalized patients at risk.

## 2. Materials and Methods

### 2.1. Study Design

This study was an investigator-initiated, prospective, single-center trial conducted at the University Hospital Basel to evaluate the feasibility, safety, performance, compliance and data availability of a PPG-based wrist-worn wearable device in combination with a device hub platform for automated data capturing and processing, in patients with no known history of atrial fibrillation (AF). If absolute arrhythmia (AA) was detected in the PPG signal, a seven-day Holter ECG was performed subsequently.

### 2.2. Sample Size

The sample size required to achieve a power of 90% to detect at least 10 patients with previously undiagnosed AF was calculated to be 353. The prevalence of AF was estimated at four percent for this calculation.

### 2.3. Ethics Approval and Patient Consent

This study was conducted in compliance with the Declaration of Helsinki, the ICH-GCP or ISO EN 14155 as well as all national legal and regulatory requirements. Ethical approval was obtained from the local ethics committee (EKNZ BASEC 2020-01983) and was registered on ClinicalTrials.gov (NCT04563572). All participants provided written informed consent.

Patients of the internal medicine ward of the University Hospital Basel were screened for inclusion and exclusion criteria based on their electronic records. Patients 18 years or older, with a CHA_2_DS_2_-VASc score of two or higher, with no known history of atrial fibrillation, without an implanted cardiac device (pacemaker or implantable cardioverter defibrillator), and with no ongoing chronic anticoagulation therapy were eligible for the study.

If all conditions were met, patients were recruited after providing informed consent. In total, 377 patients were enrolled in this study between July 2021 and July 2022. Recruitment was not restricted to a specific diagnosis; participants were admitted for a variety of conditions from the entire department of internal medicine (e.g., cardiology, oncology, pneumology, infectious diseases, nephrology and neurology). All patients were stationary, but their level of mobility and morbidity varied. Data on patients’ characteristics were obtained from electronic records and personal interviews.

### 2.4. Investigational Products

Each participant received a wristband with a built-in PPG sensor (CardioWatch 287 Bracelet, Corsano Health B. V., Bussum, The Netherlands) (Figure 1). The sensor on the wristband received the CE conformity marking. The pulse wave signals generated from the PPG data were analyzed using the Heartbeats Algorithm (Version 1.1.4) by Preventicus^®^ GmbH (Jena, Germany). The algorithm was developed for the detection and quantification of AF episodes through the analysis of the PPG signals. The algorithm has been clinically validated, is CE marked and is a certified medical device (class IIa).

The Device Hub platform by Leitwert (Zurich, Switzerland) was used to automatically capture data from the wristbands, store them on hospital servers, assign data to patients, monitor data quality and automate the exchange of deidentified data and analytics results with the Preventicus server. The Device Hub platform by Leitwert (Zurich, Switzerland) has been validated in multiple studies and pilot operations [25].

Seven-day Holter ECGs (SEER™ 1000, GE Healthcare Getemed AG, Freiburg, Germany) served as a gold standard to confirm the diagnosis subsequently (Table 2).

### 2.5. Intervention and Data Flow

Patients who met the inclusion criteria and provided informed consent received a wearable device (CardioWatch 287-1B) equipped with an integrated PPG sensor. They were instructed to wear the device continuously on their wrist during their hospital stay. The raw PPG data captured by the device were transmitted via Bluetooth to access points in the patients’ rooms. These gateways forwarded the data over the hospital’s secure WLAN to an on-site server at the University Hospital Basel. The Device Hub software monitored the continuous data flow, including signal quality, battery status, and device activity.

The deidentified PPG data, segmented into 30 min raw data packages, were automatically sent from the local server to Preventicus for analysis. The Heartbeats Algorithm processed the data and generated an ECG-comparable PDF report, which was reviewed for signs of arrhythmia, such as atrial fibrillation (AF). If AF was suspected, the PDF report was forwarded to the Telecare Center in Ulm for further verification by trained technicians. Upon completion of the quality check, the results, along with the quality status, were synchronized with the University Hospital Basel, where the study team could access the analysis and quality status (Figure 2).

The device status and data quality were continuously monitored by the study team using a front-end dashboard. This dashboard provided real-time insights into various metrics, including the online status of the Bluetooth gateway network, battery status, data synchronization completeness, and data quality (e.g., a 6 h moving average of evaluable minutes of PPG data). In cases of device issues, such as low battery or missing data due to patient noncompliance, the study team could replace devices and ensure high-quality data acquisition.

The study team used the Device Hub software to administer the study, assign devices to patient IDs, and track the status of all incoming data. If AF was suspected, the study team initiated a 7-day Holter ECG to confirm the diagnosis and asked the patient to record symptoms in a simplified Holter monitor diary. Both Preventicus and the Telecare Center Ulm had no access to any personal or clinical data beyond the deidentified 30 min segments of PPG data.

### 2.6. Algorithm

The Heartbeats Algorithm was developed by Preventicus^®^ in Jena, Germany (version 1.1.4), and was used to analyze deidentified PPG files. This algorithm was designed to differentiate between sinus rhythm (SR) and absolute arrhythmia (AA) using a complex, non-linear combination analysis of beat-to-beat changes in pulse wave time intervals and pulse wave morphology [14,15]. The Heartbeats Algorithm assessed 60 consecutive seconds of data to classify the one-minute segment as “AA”, “SR”, or “Noise” (e.g., insufficient quality). Before conducting rhythm analysis, the algorithm checked the quality of the data by examining the signal-to-noise ratio and accelerometer data. Data segments with excessive noise mainly due to movement artefacts were automatically detected and excluded from the analysis. If the percentage of noise or disturbed data segments compared to good quality data exceeded a threshold of 5% for AA segments or 30% for SR segments, these segments were rejected. As an example, if more than 18 s of noise was present in a 1-min segment of SR, this 1-min segment was labeled noise. Meanwhile, if more than 3 s of noise was present in a 1-min segment of AA, the segment was excluded. Only segments with AA lasting at least two consecutive minutes were considered as AA. The evaluable monitoring time was defined as the sum of the time that was categorized as either SR or AA, meaning all segments not labeled as noise.

To verify suspected AA episodes, trained telecare platform technicians performed PPG quality control and conducted an in-depth analysis including accelerometer data to determine if AA was plausible. Confirmed AA episodes were documented with a start and end timestamp and labeled as “AA”. After evaluation, the recordings were automatically converted into an ECG-comparable report. This process is the standard procedure of the software company, and it is certified as a medical device.

### 2.7. Terminology

In this study, the term absolute arrhythmia (AA) was used to describe irregular pulse wave patterns detected by the PPG-based algorithm that were suspicious for atrial fibrillation but not yet confirmed via ECG. AA served as a screening label, indicating a potentially pathological rhythm requiring further evaluation. It did not equate to a definitive diagnosis of atrial fibrillation but rather represented a flagged event suggestive of AF based on pulse irregularity.

### 2.8. Interpretation of ECG

A supervising physician and a senior cardiologist at the University Hospital Basel analyzed the ECG recordings independently, using standard software (Cardioday^®^ V2.5, GE Healthcare, Getemed AG, Freiburg, Germany). The ECG was first automatically analyzed by the software. The supervising physician then reviewed the analysis and determined whether the automated interpretation was accurate or needed to be adjusted. A senior cardiologist provided supervision and reviewed all recordings, classifying certain segments as either “AF” or “Noise”. Atrial flutter was labeled as AF, because it has the same therapeutic consequences according to the ESC guidelines [6]. An episode of AF was defined as absolute atrial arrhythmia lasting more than 1 min, rather than the more common definition of at least 30 s [6]. This modification was implemented, because the Heartbeats Algorithm requires a minimum of one minute to classify a segment accordingly.

### 2.9. Outcomes

The primary outcomes of the trial were the overall success of automatic data transmission, analysis and visualization, the number of identified study subjects with AA by using the Heartbeats Algorithm and a PPG sensor, and the number of subsequent Holter ECGs. Secondary outcomes included the screening time until first detection of AA, the distribution of AA burden throughout the day, compliance with wearing the PPG device, and an assessment of the automatic analysis process by analyzing data availability from the wearable device.

### 2.10. Compliance

A patient was defined as compliant if at least one evaluable minute was received by the Device Hub for every day during their hospitalization or until transfer to a different unit. A patient was defined as noncompliant if no evaluable time was received on at least one day of their hospitalization period or if the trial was terminated prematurely. In cases of premature study termination or data suggesting noncompliance, the reasons were assessed retrospectively.

### 2.11. Follow-Up and Adverse Events

A follow-up was conducted for all participants who terminated the study irregularly or if the data indicated noncompliance at the end of the study to assess the reason. The follow-up was performed by means of personal interview or telephone contact. All patients with AA in the PPG signal were visited to record a seven-day Holter ECG, and the European Heart Rhythm Association (EHRA) score was evaluated.

Adverse events were assessed at the end of the study by means of personal interview if possible. All patients were screened for AF, TIA, stroke and thromboembolic insults based on the electronic patient record at the end of the study.

### 2.12. Statistical Analysis

Continuous values are expressed as the mean and standard deviation; categorical values are expressed as absolute numbers and percentages. Continuous values were tested for normality using the Shapiro–Wilk test. Based on the results of this, either the *t*-test or the Wilcoxon rank sum test was employed. Categorical variables were analyzed using the chi-squared test; in cases of a small sample size, the Fisher exact test was used to determine any significant differences. The prevalence of AA including standard error within the 95% confidence interval (CI) was calculated. Statistical significance was determined by performing two-sided tests and setting the threshold for the *p*-value at 0.05. A *p*-value less than 0.05 was considered to indicate a statistically significant relationship between the variables. All statistical analyses were performed in RStudio (Version R 4.2.2).

## 3. Results and Discussion

### 3.1. Patient Characteristics

A total of 377 patients were enrolled between July 2021 and July 2022. Thirty-one patients had to be excluded from the analysis for various reasons (Figure 3). A total of 346 patients were included for data analysis. Of those, twelve had AA in the PPG-derived data.

In the analyzed population, the median age was 73, ranging from 32 to 96. Overall, 57 patients were younger than 65, while 145 were 75 or older. There were 175 (50.6%) females and 171 (49.4%) males. The mean CHA_2_DS_2_-VASC score was 3.5 ± 1.3, while the median score was 3, ranging from 2 to 9. The most frequent condition from the CHA_2_DS_2_-VASc score was hypertension (63.6%) followed by vascular disease (41.6%), not including the age and sex categories. There were no statistically significant differences between the two groups with AA and without AA (Table 3). An extended table with more detailed information including various baseline diagnoses and medications can be found in the Appendix A.

### 3.2. PPG Analysis

A total of 2’141’219 min (4.1 years) of data was collected for the 346 patients. Of those, 982’086 min (1.9 years) was evaluable monitoring time (45.9%). On average, each patient wore the wristband for 6186 ± 6280 min (4.3 ± 4.4 days), of which 2838 ± 3180 min (2.0 ± 2.2 days) was evaluable monitoring time. The median wearing duration was 4479 min (3.1 days), ranging from 43 min to 69’355 min (48.16 days). All data were transmitted, analyzed, and stored, and no loss of data occurred in the process. In 12 patients (3.5%, CI: 1.5–5.4%), AA was noted in the PPG data. A combined total of 4005 min (2.8 days) was identified as AA in these patients. This made up 14.2% of the total evaluable time. These 12 patients wore the wristband for 5343 ± 3438 min (3.7 ± 2.4 days) on average; the median time was 4550 min (3.2 days), ranging from 1138 min (0.8 days) to 14’415 min (10.0 days). The evaluable time in these 12 patients was, on average, 44%. The median evaluable monitoring time was 2177 min (1.5 days), ranging from 39 min to 5220 min (3.6 days).

In the same group of patients, the mean AA burden was 334 ± 449 min (5.6 ± 7.5 h) per patient, while the median was 126 min (2.1 h), ranging from 5 min to 1418 min (1.0 days). The longest individual AA episode lasted, on average, 14.8 ± 13.2 min, ranging from 2 min to 39 min. There were 1326 individual AA episodes, averaging at 111 ± 138 individual AA episodes per patient, and the median was 39.5, ranging from 3 to 432. On average, each AA episode lasted 3.0 min. The mean time from the beginning of screening until the first detection of AA was 883 ± 1009 min (14.7 ± 16.8 h), while the median was 378 min (6.3 h), ranging from 1.2 h to 54.7 h (2.3 days). The incidence rate was higher in the first 24 h of wearing the wristband compared to the remaining time (*p* = 0.004) and the following 24 h (*p* = 0.012) (Figure 4).

### 3.3. ECG Analysis

In 12 patients, the PPG analysis indicated at least two consecutive minutes of AA. To confirm AF, Holter ECG recordings were started a median of 3.1 days (IQR 1.2 days) after the first detection of AA. Nine of the twelve patients underwent a seven-day Holter ECG subsequently, while three refused to receive one. In these nine patients, a total of 55’197 min (38.3 days) of data was recorded. The mean duration was 6133 ± 4343 min (4.3 ± 3.0 days), while the median was 8846 min (5.8 days), ranging from 9 min to 10’080 min (7 days). There were no AF segments in the analyzed ECG recordings, and no symptomatic episodes were recorded in the simplified Holter diary.

### 3.4. Compliance

Overall, 304 of the 346 patients were compliant (87.9%) and 42 were noncompliant (12.1%). Twelve patients were initially marked as noncompliant but later revised to compliant as the reason was solely technical (e.g., battery was not recharged in time by study staff). Twenty participants terminated the study prematurely. Various reasons for noncompliance were assessed (Table 4). No rashes or allergic reactions from wearing the device were documented.

The baseline characteristics of the two groups (compliant and noncompliant patients) were analyzed. No statistically significant differences were identified between the two populations of compliant and noncompliant patients (Appendix A).

### 3.5. Adverse Events and Deaths

No adverse events were recorded. There were no reported strokes, TIA, or thromboembolic insults at the end of study. Two patients died by the follow-up period, which was conducted after the end of study. The deaths occurred two and three months after the termination of the regular study, and both deaths were unrelated to the study procedures.

### 3.6. Discussion

This trial successfully developed and deployed an IT infrastructure for fully automated, continuous AF screening in hospitalized patients using wrist-worn PPG-based wearable devices. To the authors’ knowledge, this the first hospital implementation of a fully automated system for AF screening based on PPG technology.

The key findings of this study highlight the technical feasibility and operational success of this innovative system:The automated screening infrastructure functioned reliably, ensuring continuous data acquisition, processing, and visualization. No data were lost, demonstrating seamless system integration within a hospital environment.The system identified absolute arrhythmias in 3.5% of hospitalized patients, showcasing the potential of automated real-time monitoring.High patient acceptance was observed, with nearly 90% compliance.Automated analysis revealed that arrhythmic burden was 1.3 times higher at night, emphasizing the advantage of continuous tracking, particularly at night.

This study demonstrates the feasibility of implementing a hospital-wide, automated absolute arrythmia detection system using wearables, a structured data pipeline, and real-time visualization. The visualized data on the dashboard enabled quick insights and possibly led to high compliance rates and high signal integrity. Common problems with wrist-worn PPG sensors like the battery life or no proper contact between the sensor and skin, due to improper wearing of the device, leading to low signal quality, could be minimized in this process.

Approximately half (46%) of the time the PPG signal was good enough for analysis, which is comparable to other studies with wrist-worn PPG sensors [26,27].

Furthermore, systematic screening for atrial fibrillation using this algorithm was demonstrated to be cost-effective in multiple countries in Europe, and quality-adjusted life years can be gained [28,29]. In conclusion, this system could offer a scalable, cost-effective solution for a hospital environment, setting a precedent for future digital health innovations in inpatient care.

The AA detection rate of 3.5% is comparable to other studies, with a similar design. In the Smart in OAC study conducted by Fabritz et al. [26], a 5% incidence was detected, likely because the study population was 65 years or older and the screening duration was longer (four to eight weeks). The Apple Heart Study [17], the Huawei Heart Study [23], and the Fitbit Heart Study [19] had lower AA incidences ranging from 0.23% to 1.0%, which is likely due to the younger population with fewer risk factors at baseline. In contrast, higher AA rates (10.1%, 20.7% and 31.8%) were identified when screening was performed with implantable loop recorders in high-risk patients [30,31,32].

Out of nine AA cases that were effectively followed up with a Holter ECG, no cases of AF were confirmed. Previous studies have reported confirmation rates between 32% and 87% [17,18,23,26]. The low confirmation rate in our study could be explained by the small sample size of AA and the inclusion of AA lasting only two minutes, instead of a minimum of six minutes. We chose two minutes because unlike six-minute or longer segments, which are known to be associated with an increased risk of ischemic stroke [33,34], the clinical relevance of very brief sporadic AF episodes lasting under six minutes is still under investigation [35]. However, it is possible that this lower threshold may have led to an increased number of false-positive episodes. Other studies using a six-minute threshold have reported higher confirmation rates, suggesting that threshold duration may significantly influence the diagnostic accuracy. Another explanation could be that some ECG recordings were very short (two of the nine were less than one hour, instead of seven days) and started with a delay of about three days. In the analysis of the ECG data, multiple instances of sinus arrhythmia, premature atrial contractions, and premature ventricle contractions occurred. These misclassifications are a recognized limitation to PPG-based algorithms, possibly contributing to the apparent high false-positive results in this study [21,36]. They could further explain the lack of AF confirmations. Previously, the same algorithm that was used in this trial demonstrated very high accuracy in detecting AF [13,26,27]. The presence of these benign arrhythmias underscores the importance of the continued refinement of PPG-based algorithms to reduce false positives and improve diagnostic accuracy.

Despite the technical success of this study, the short average PPG monitoring time of 4.3 days, with a median of 5.8 days, may not be sufficient to rule out paroxysmal AF. Previous studies have demonstrated higher confirmation rates of AF with extended screening periods [26]. Similarly higher detection rates could be identified using loop recorders that allowed for long-term monitoring [30,31,32]. In our study, the limited monitoring duration of both PPG wearable and the subsequent Holter ECG likely contributed to the absence of confirmed AF cases. To strengthen diagnostic accuracy, future implementations should consider initiating PPG monitoring immediately upon hospital admission and extending it beyond discharge. This could increase the chance of capturing paroxysmal AF episodes and improve the robustness of a hospital-based AF screening infrastructure.

Short screening durations of 24 h were enough to find the majority of AA cases in a mean screening duration of 4 days. The Smart in OAC study had the highest detection rate in the first week of a four-week screening cycle [26]. Both studies displayed peak detection in the first quarter of the screening period. The difference between the results from one day to seven days can be explained by the inclusion of AA segments lasting at least two minutes versus six minutes. This highlights the need for future studies to determine the optimal monitoring duration and AA episode threshold length based on individual patient risk profiles. Tailoring these parameters could improve diagnostic yield while minimizing false positives, thereby enhancing the clinical utility of hospital-based AF screening programs.

The majority of the AA burden was identified in the night, particularly from 03:00 to 03:59 and from 22:00 to 22:59 in the night. This is aligned with the findings of Deguchi et al., who demonstrated that paroxysmal AF episodes in Holter ECGs occurred most frequently around midnight [37]. Ultimately, this encourages continuous AF screening, including monitoring during the night. We see this infrastructure for automatic AF screening in hospitalized patients as a complementary modality to existing AF screening methods. It should not be seen as a competing method. Hospitalized patients at risk for AF will still need a long-term Holter ECG or implantable loop recorder to confirm the diagnosis. However, this cannot cover all patients at risk. The solution provided in this study could potentially fill this gap between external long-term ECG and implantable recorders. The visualized data allow for the convenient administration of multiple patients simultaneously, and with timely analysis, they enable therapeutic decisions with minimal delay.

### 3.7. Limitations

Our study had several limitations. A known limitation of all PPG-based devices is their high susceptibility to artefacts, often caused by motion, which lead to noise and unusable data [22]. Although factors such as vasoconstriction, fluid shift, or medication effects were not systematically analyzed, they may have contributed to reduced signal quality in some patients. False positives due to sinus arrhythmia and premature atrial and ventricular contractions remain an issue, which is why current guidelines still require a confirmatory ECG to make an AF diagnosis. Beyond the scope of this study, wearable biosensors face several fundamental technological challenges that may affect their reliability in prolonged clinical use. These include issues related to signal stability, power consumption, and secure data transmission, which are critical for effective and scalable implementation in healthcare settings [38]. While these issues were not prominent in our study setting, they may affect performance in other clinical environments. Addressing these factors and continuously optimizing them will be essential to ensure seamless functionality and a robust wearable-based infrastructure in clinical settings.

The study design required subsequent Holter ECGs after detecting AA in the PPG signal in order to confirm AF. However, it proved challenging to convince participants to undergo a follow-up Holter ECG and wear it for the full 7-day period, which could have further limited the diagnostic yield. The paroxysmal nature of atrial fibrillation and the absence of simultaneous recordings using PPG and ECG limited our ability to confirm whether arrhythmias detected by the algorithm corresponded to true AF episodes, thereby reducing diagnostic accuracy. Retrospectively, it was not possible to rule out AF entirely. Moreover, the relatively short monitoring time of the wearable device, combined with the delay in initiating the Holter ECG, may have led to missed episodes. In some cases, the Holter ECG recordings lasted for less than one hour, limiting their diagnostic utility.

The reasons for noncompliance were assessed in retrospect, which could have led to recall bias. It is important to note that while recall bias is a potential limitation, the low noncompliance rate suggests that this limitation may have only had a minimal impact on the overall findings. Furthermore, reasons for noncompliance such as the implantation of pacemakers or the initiation of Holter ECGs can be verified objectively and further strengthen the validity of the results despite the retrospective approach.

Lastly, the statistical comparison between the groups with AA and without AA should be interpreted with caution due to the low number of AA cases (*n* = 12). This limited sample size significantly reduces the statistical power, making subgroup analyses unreliable, and thus may not accurately reflect the true underlying relationship. Further investigations with larger cohorts are needed to confirm the observed trends.

### 3.8. Conclusions

We have successfully built an innovative infrastructure within the University Hospital Basel that allows for the real-time visualization and monitoring of multiple hospitalized patients simultaneously for AF. An automatic screening method using PPG-based wearables identified absolute arrhythmias in 3.5% of an at-risk hospitalized population with no history of atrial fibrillation. Most absolute arrhythmias were detected during the night and in the first 24 h of screening. These findings highlight the potential of integrating wearable-based monitoring into clinical workflows to improve early AF detection. Optimizing monitoring durations, ensuring timely confirmatory ECG and refining detection thresholds and algorithms may enhance diagnostic accuracy and clinical utility. Future studies are needed to evaluate clinical outcomes and cost-effectiveness on a larger scale. Ultimately, this infrastructure could serve as a blueprint for scalable, hospital-wide AF screening programs.

## Figures and Tables

**Figure 1 diagnostics-15-01233-f001:**
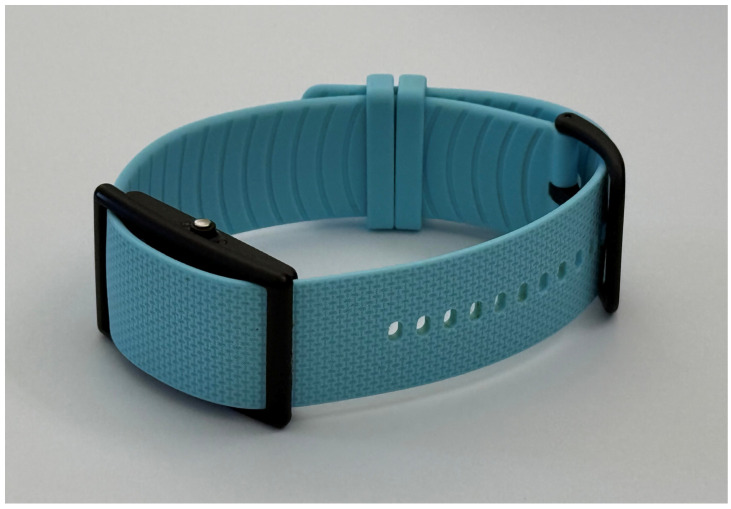
Wristband with integrated PPG sensor.

**Figure 2 diagnostics-15-01233-f002:**
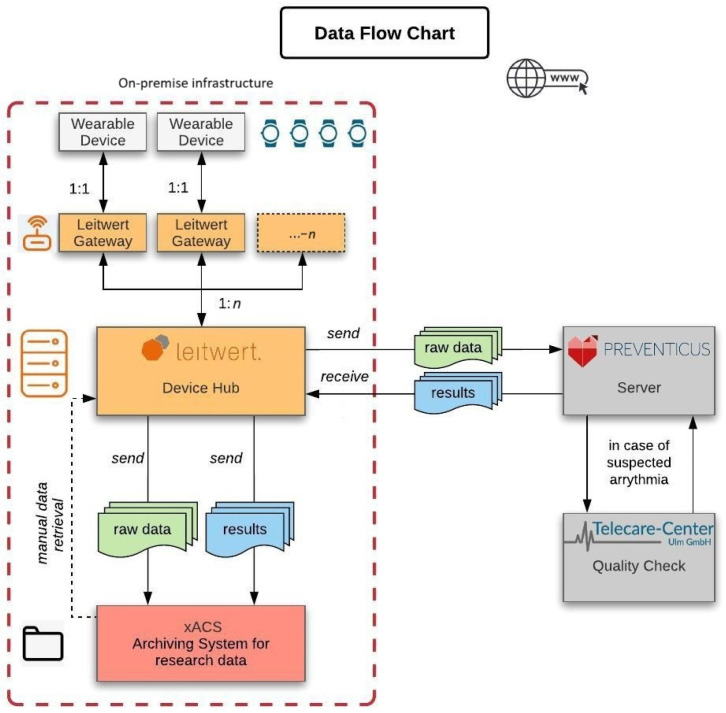
Data flowchart. Data from the wearable device were continuously collected and transmitted via Bluetooth to a local gateway. Each wearable device was paired with a dedicated gateway (1:1). The gateway then forwarded the data securely over WLAN to a server, deployed on-site at the University Hospital Basel (UHBS). The server (Device Hub) maintained communication with multiple gateways (1:*n*) to monitor data flow, signal quality, and battery status. The deidentified raw PPG data were then sent from the server to an external server of Preventicus for analysis, where they were evaluated for conditions such as atrial fibrillation (AF), sinus rhythm (SR), or noise. An ECG-comparable PDF report was generated based on the results. In cases of suspected arrhythmia, the data were forwarded to the Telecare Center Ulm for further quality control and confirmation by trained technicians. The results of the analysis, along with the status of the quality control, were continuously synchronized with the server at the University Hospital Basel and re-identified with the patient ID for tracking and clinical reference.

**Figure 3 diagnostics-15-01233-f003:**
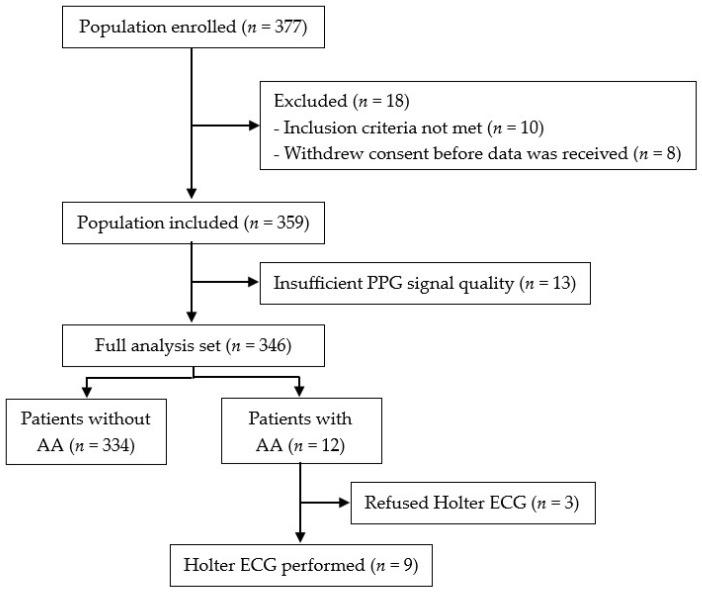
Flowchart of the study inclusions, exclusions and full analysis set. AA, absolute arrhythmia; ECG, electrocardiography; PPG, photoplethysmography.

**Figure 4 diagnostics-15-01233-f004:**
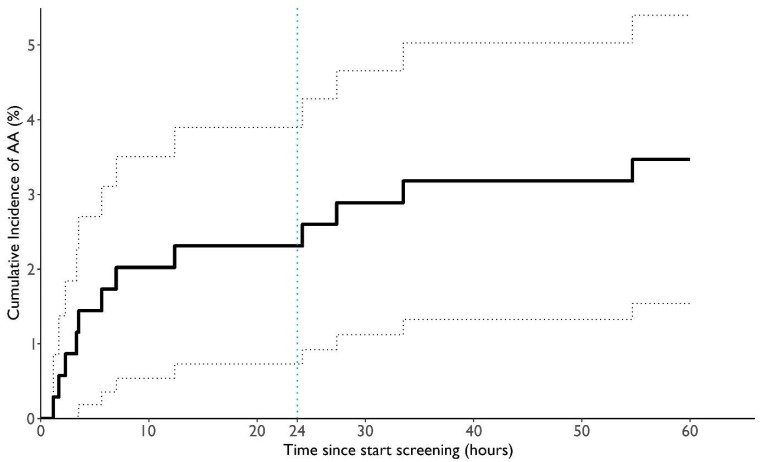
Time to first detection of AA in the PPG data. The black line indicates the Kaplan–Meier estimated cumulative incidence rate of first AA detection. The dotted black lines show the corresponding 95% confidence interval. The vertical dotted blue line indicates the 24 h mark. The total AA burden per hour from all patients with detected AA was examined (Figure 5). The AA burden per hour was 1.3 times higher during the night (22:00–06:00) compared to the day (*p* = 0.03) (corrected for better signal quality at night). The highest AA burden occurred from 03:00 to 03:59 in the night, followed by between 22:00 and 22:59. The lowest AA burden was observed from 16:00 to 16:59 in the afternoon.

**Figure 5 diagnostics-15-01233-f005:**
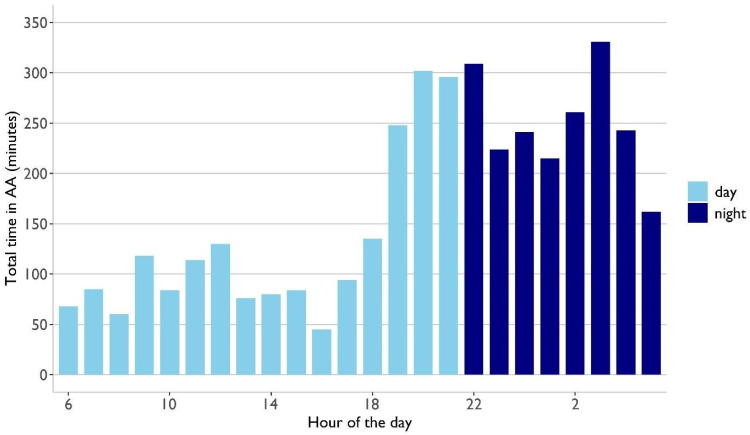
Total AA burden per hour. Each bar represents the total time spent in AA within one hour from all patients with AA. For example, at hour 6, lasting from 06:00 to 06:59, 68 min of AA was identified. The bright blue bars are time intervals during the day and dark blue bars are time intervals during the night (22:00–06:00). The two hours with the greatest AA burden occurred at night.

**Table 1 diagnostics-15-01233-t001:** Comparison of different algorithms based on PPG data.

Manufacturer	Device	Sensitivity (%)	Specificity (%)	Positive Predictive Value (%)	Reference
Preventicus	Device agnostic: Samsung Gear Fit II, Corsano Cardiowatch 287-1, Polar Verity Sense	93.7 (CI: 89.8–96.4)	98.2 (CI: 95.8–99.4)	97.8 (CI: 94.9–99.3)	[13]
Apple	Apple Watch 2	Not applicable	Not applicable	84.0 (CI: 76.0–92.0)	[17]
Fitbit	Fitbit devices	67.6 (CI: 62.4–72.6)	98.4 (CI: 97.3–99.2)	98.2 (CI: 95.5–99.5)	[22]
Huawei	Huawei Watch GT, Honor Watch	Not applicable	Not applicable	91.6 (CI: 91.5–91.8)	[23]
Samsung	Samsung Galaxy Watch Active 2	87.8 (CI: 83.6–91.0)	97.4 (CI: 97.1–97.7)	Not applicable	[24]

**Table 2 diagnostics-15-01233-t002:** Overview of products used in this study.

Product	Vendor/Make	Model/Version	Country
Wristband with PPG Sensor	Corsano Health B.V.	CardioWatch 287 Bracelet	The Netherlands
AF Detection Algorithm	Preventicus GmbH	Heartbeats Algorithm v1.1.4	Germany
Device Hub Platform	Leitwert AG	Device Hub Platform	Switzerland
Holter ECG Device	GE Healthcare Getemed AG	SEER^TM^ 1000	Germany

**Table 3 diagnostics-15-01233-t003:** Characteristics of all study participants, grouped into those with and without absolute arrhythmia.

		Total (*n* = 346)	Without AA (*n* = 334)	With AA (*n* = 12)	*p*-Value
Age (years)	Mean ± SD	72.0 ± 9.8	71.8 ± 9.8	76.9 (10.2)	
	Median (IQR)	73 (12)	72.5 (12)	79.5 (11.5)	0.63 ^1^
	Range	32–96	32–96	58–89	
Sex	Female	175 (50.6%)	170 (50.9%)	5 (41.7%)	0.738 ^2^
	Male	171 (49.4%)	164 (49.1%)	7 (58.3%)	
BMI	Mean ± SD	26.7 ± 5.5	26.8 ± 5.5	26.0 ± 5.2	0.708 ^1^
CHA_2_DS_2_-VASc score	Mean ± SD	3.5 ± 1.3	3.5 ± 1.3	4.0 ± 1.5	0.207 ^1^
	Median (IQR)	3 (2)	3 (2)	3.5 (2.25)	
	Range	2–9	2–9	2–6	
Tattoo in sensor area	Yes	1 (0.3%)	1 (0.3%)	0	-
Skin color Fitzpatrick scale ^3^	Mean ± SD	2.2 ± 0.7	2.2 ± 0.7	2.2 ± 0.6	0.908 ^1^

^1^ Wilcoxon rank sum test; ^2^ chi-squared test; ^3^ from one to six (pale white to dark brown/black). BMI, body mass index; SD, standard deviation; IQR, interquartile range.

**Table 4 diagnostics-15-01233-t004:** Reasons for noncompliance.

		Noncompliant Patients (*n* = 42)
Irregular study termination (*n* = 20)	Not convinced by the wristband	6
	Reason not provided	3
	Pacemaker or Holter	4
	Afraid of PPG sensor	2
	Health-related (unrelated to the study)	5
Regular study termination (*n* = 22)	Forgot to wear device	6
	Unaware device had to be worn at all times	4
	Uncomfortable	5
	Itchiness	3
	Lost to follow-up	2
	Deceased	2

The table is divided into two categories, the first being patients that terminated the study prematurely and the second being patients that had finished the study as per the protocol but the data indicated noncompliance in retrospect. The reasons were assessed in retrospect.

## Data Availability

The data underlying this article will be shared upon reasonable request to the corresponding author.

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
