# Peer review of "Fully Automated Photoplethysmography-Based Wearable Atrial Fibrillation Screening in a Hospital Setting"

_diagnostics, 2025, doi:10.3390/diagnostics15101233_

Round 1
Reviewer 1 Report
Comments and Suggestions for Authors
The authors developed a continuous monitoring system for Atrial Fibrillation in a hospital setting. The study is interesting and presented well. My comments are
- The mean monitoring duration of 4.3 days and a median of 5.8 days is relatively short to confirm the lack of Atrial Fibrillation. Extending the Holter ECG monitoring period can improve validation.
- Since none of the Holter ECG recordings confirmed AF. This may indicate a high false-positive rate in the PPG algorithm. Authors can discuss the potential impact of premature atrial contractions and other arrhythmias on misclassification
- A comparison table showing the sensitivity and specificity of the Preventicus Heartbeats Algorithm with other validated PPG-based AF detection methods will strengthen the technical evaluation.
- Hospitalized patients may have altered physiological states like fluid shifts, medication effects, etc… will these factors influence the accuracy of PPG-based AF detection?
- Are there any cases where simultaneous PPG and ECG recordings were taken?
- In this study, a two-minute threshold is used for absolute arrhythmia classification. Whereas other studies used a six-minute criterion. Does a lower threshold lead to excessive false positives and does adjusting threshold have any significance in improving diagnosis?
Author Response
"Please see the attachment"

Reviewer 2 Report
Comments and Suggestions for Authors
This study addresses a timely and clinically significant topic—automated atrial fibrillation (AF) screening using wearable photoplethysmography (PPG) technology. The integration of PPG-based wearables into hospital workflows is a novel and promising concept, and the study effectively demonstrates its feasibility.
Major Comments:
- Methods: Please provide details on the reasons for hospitalization of the study participants. Additionally, clarify their ambulatory status, as this may influence the applicability of PPG-based monitoring.
- Figure 1: The notation “1:n” is unclear; please clarify its meaning. Additionally, correct the wording of “receive” to ensure grammatical accuracy.
- Terminology: The distinction between absolute arrhythmia (AA) and atrial fibrillation should be explicitly explained in the text. Given that AA was used as a screening criterion, a clearer definition would aid in understanding its clinical significance.
- Subgroup Analysis: The small number of AA cases (n=12) limits the feasibility of subgroup analyses. The lack of statistical power in comparisons between AA-positive and AA-negative groups should be acknowledged in the limitations section.
- False Positives: The study detected absolute arrhythmia (AA) in 3.5% of participants, yet none of the subsequent Holter ECGs confirmed AF. This raises concerns about potential false positives in PPG-based detection, which should be discussed in the context of clinical implications and possible refinements in detection algorithms.
- Figure Legends: The figure legends could be clearer. For instance, Figure 3 should explicitly state what the Kaplan-Meier curve represents to ensure better reader comprehension.
7.Device Presentation: Including a photograph of the wrist-worn PPG device would enhance clarity regarding the technology used in the study.
Comments on the Quality of English Languageplease refine.
Reviewer 3 Report
Comments and Suggestions for Authors
The authors investigated a clinical trial conducted at the University Hospital Basel to assess a PPG-based wrist-worn wearable device coupled to an automated data processing hub for AF screening in hospitalized patients for real-time application. The article is poorly written and does not cover the key factors compared with the most recent state-of-the-art. There are several flaws and errors that need to be addressed. It is unclear on the novelty of the proposed work. Below are the comments:
1) The title should be rephrased in a more meaningful way
2) Instead of background/objectives, add an abstract of 250 words
3) Introduction is too short and needs to be discussed comprehensively with key parameters such as wearable devices, data analysis
4) Authors can add these relevant references to support their work: 10.1016/j.talanta.2024.125817; 10.1109/JPROC.2022.3149785
5) In section 2, add the consumables table, including vendor, make, model, country
6) In Figure 1, name the small box within the dash lines
7) Figure 2 resolution should be increased
8) Merge results and discussion sections
9) What are the limitations and challenges
10) Add the most recent references
11) The English language should be improvised and corrected
Comments on the Quality of English LanguageThe authors investigated a clinical trial conducted at the University Hospital Basel to assess a PPG-based wrist-worn wearable device coupled to an automated data processing hub for AF screening in hospitalized patients for real-time application. The article is poorly written and does not cover the key factors compared with the most recent state-of-the-art. There are several flaws and errors that need to be addressed. It is unclear on the novelty of the proposed work. Below are the comments:
1) The title should be rephrased in a more meaningful way
2) Instead of background/objectives, add an abstract of 250 words
3) Introduction is too short and needs to be discussed comprehensively with key parameters such as wearable devices, data analysis
4) Authors can add these relevant references to support their work: 10.1016/j.talanta.2024.125817; 10.1109/JPROC.2022.3149785
5) In section 2, add the consumables table, including vendor, make, model, country
6) In Figure 1, name the small box within the dash lines
7) Figure 2 resolution should be increased
8) Merge results and discussion sections
9) What are the limitations and challenges
10) Add the most recent references
11) The English language should be improvised and corrected
Round 2
Reviewer 3 Report
Comments and Suggestions for Authors
Authors have made significant changes to the revised version; however, no paper exists that refers to reference 38. Authors need to add the correct one and update it. Please check the previous comments for details.
" [38] Kulkarni MM, Patil S, Kulkarni RM, Aland RC, Nayak MM. Advances in Smart Wearable Biosensors for Non-Invasive Health 683
Monitoring: A Comprehensive Review. Talanta. 2024 Apr 1;269:125817" - Does not exist?
Author Response
Response to Reviewer 3 Comments
We sincerely thank you for taking the time again to evaluate our revised manuscript.
Comment 1:
Authors have made significant changes to the revised version; however, no paper exists that refers to reference 38. Authors need to add the correct one and update it. Please check the previous comments for details.
" [38] Kulkarni MM, Patil S, Kulkarni RM, Aland RC, Nayak MM. Advances in Smart Wearable Biosensors for Non-Invasive Health 683
Monitoring: A Comprehensive Review. Talanta. 2024 Apr 1;269:125817" - Does not exist?"
Response 1:
We sincerely thank the reviewer for identifying this critical issue and for suggesting the inclusion of this valuable publication. Upon review, we recognized that reference [38] was indeed incorrectly cited due to a referencing software error. We regret this oversight, which occurred despite our quality control procedures.
The intended reference was:
Kulkarni MB, Rajagopal S, Prieto-Simón B, Pogue BW. Recent advances in smart wearable sensors for continuous human health monitoring. Talanta. 2024 May 15;272:125817.
This reference has now been correctly cited in the revised manuscript, additionally we have thoroughly reviewed and corrected the entire reference list. Lastly, we have slightly adjusted the corresponding paragraph to better reflect the scope and findings of this comprehensive review article.
We greatly appreciate the reviewer’s careful reading and the suggestion to incorporate this recent and highly relevant publication. It has significantly strengthened the scientific foundation of our manuscript and has contributed to a more robust discussion of the technological challenges associated with wearable biosensors in clinical environments.